# Necroptosis Mediates Muscle Protein Degradation in a Cachexia Model of Weanling Pig with Lipopolysaccharide Challenge

**DOI:** 10.3390/ijms241310923

**Published:** 2023-06-30

**Authors:** Junjie Guo, Xu Qin, Yang Wang, Xiangen Li, Xiuying Wang, Huiling Zhu, Shaokui Chen, Jiangchao Zhao, Kan Xiao, Yulan Liu

**Affiliations:** 1Hubei Key Laboratory of Animal Nutrition and Feed Science, Wuhan Polytechnic University, Wuhan 430023, China; dkxyyjs123@126.com (J.G.); 19356055812@163.com (X.Q.); 15907185411@163.com (Y.W.); genuncle@163.com (X.L.); x.wang@igbzpan.pl (X.W.); zhuhuiling2004@sina.com (H.Z.); loveskchen@163.com (S.C.); 2Department of Animal Science, Division of Agriculture, University of Arkansas, Fayetteville, AR 72701, USA; jzhao77@uark.edu; 3School of Animal Science and Nutritional Engineering, Wuhan Polytechnic University, No. 68 Xuefu South Rd., Wuhan 430023, China

**Keywords:** necroptosis, cachexia, pro-inflammatory cytokines, protein degradation

## Abstract

Necroptosis, an actively researched form of programmed cell death closely related to the inflammatory response, is important in a variety of disorders and diseases. However, the relationship between necroptosis and muscle protein degradation in cachexia is rarely reported. This study aimed to elucidate whether necroptosis played a crucial role in muscle protein degradation in a cachexia model of weaned piglets induced by lipopolysaccharide (LPS). In Experiment 1, the piglets were intraperitoneally injected with LPS to construct the cachexia model, and sacrificed at different time points after LPS injection (1, 2, 4, 8, 12, and 24 h). In Experiment 2, necrostatin-1 (Nec-1), a necroptosis blocker, was pretreated in piglets before the injection of LPS to inhibit the occurrence of necroptosis. Blood and longissimus dorsi muscle samples were collected for further analysis. In the piglet model with LPS-induced cachexia, the morphological and ultrastructural damage, and the release of pro-inflammatory cytokines including tumor necrosis factor (*TNF*)*-α*, interleukin (*IL*)*-1β*, and *IL-6* were dynamically elicited in longissimus dorsi muscle. Further, protein concentration and protein/DNA ratio were dynamically decreased, and protein degradation signaling pathway, containing serine/threonine kinase (*Akt*), Forkhead box O (*FOXO*), muscular atrophy F-box (*MAFbx*), and muscle ring finger protein 1 (MuRF1), was dynamically activated in piglets after LPS challenge. Moreover, mRNA and protein expression of necroptosis signals including receptor-interacting protein kinase (*RIP*)*1*, *RIP3*, and mixed lineage kinase domain-like pseudokinase (*MLKL*), were time-independently upregulated. Subsequently, when Nec-1 was used to inhibit necroptosis, the morphological damage, the increase in expression of pro-inflammatory cytokines, the reduction in protein content and protein/DNA ratio, and the activation of the protein degradation signaling pathway were alleviated. These results provide the first evidence that necroptosis mediates muscle protein degradation in cachexia by LPS challenge.

## 1. Introduction

Cachexia, a complex clinical disease, is characterized by anorexia, emaciation, and muscle atrophy [1]. Specifically, cachexia leads to reducing protein synthesis and enhancing degradation, which declines the quality, strength, and function of skeletal muscle in patients [2]. Cancer cachexia is now acknowledged to not only be induced by malnutrition, but also caused by the abnormalities of metabolism, and immune and nerve responses, of which proinflammatory factors play a detrimental role in promoting the development of cachexia [3,4]. While its mechanism of pathogenesis has been widely investigated, the molecular basis of cachexia affecting muscle protein degradation remains elusive.

The serine/threonine kinase (Akt)/Forkhead box O (FOXO) is now considered to be one of the main pathways for regulating muscle protein degradation. FOXO interacts with the ubiquitin-proteasome system (UPS) to activate its key genes, muscular atrophy F-box (MAFbx) and muscle ring finger protein 1 (MuRF1) [5,6]. Importantly, MAFbx and MuRF1 are key substances for triggering muscle atrophy and protein degradation [7,8]. Furthermore, proinflammatory cytokines such as tumor necrosis factor (TNF)-α, interleukin (IL)-1β, and IL-6 can activate the Akt/FOXO/UPS signaling pathway and are considered to be deeply involved in muscle protein degradation [9].

Previous evidence suggests that necrosis and apoptosis are two main cell death models playing an important role in muscle atrophy and regeneration in various acute and chronic musculoskeletal disorders including cachexia, sarcopenia, and aging [10,11,12]. Recently, necroptosis, a new pattern of programmed cell death, has been discovered, which possesses the same morphological characteristics as necrosis and is strictly regulated by signaling molecules similar to apoptosis [13,14]. Convincing evidence has shown that the induction of necroptosis by TNF-α is a canonical pathway [15]. Briefly, receptor-interacting protein kinase (RIP)1 is autophosphorylated under the inhibition of caspase-8, and then RIP3 and mixed lineage kinase domain-like pseudokinase (MLKL) are phosphorylated to form a necrotic complex, which is known as a key mechanism controlling necroptosis [16]. As necroptosis progresses, cells rupture and release a lot of cellular contents, which leads to inflammatory reactions and secondary injury of nearby tissues [17].

Accumulating research has suggested that necroptosis plays an essential role in multiple diseases of various tissues and organs, such as dextran sulfate sodium-induced colitis [18], non-alcoholic fatty liver disease [19], and arsenic-induced neuronal necroptosis [20]. However, the role of necroptosis in cachexia-induced muscle protein degradation has not been investigated. Thus, in this study, we aimed to investigate if necroptosis might play an essential role in muscle protein degradation induced by cachexia. The pig is thought of as an ideal model animal for studying various physiological and pathological disorders of humans [21]. Here, we constructed a piglet model of cachexia by challenging them with lipopolysaccharide (LPS) [22], the important component of the outer membrane of Gram-negative bacteria, to investigate the relationship of muscle protein degradation and necroptosis. Further, necrostatin-1 (Nec-1), a frequently used inhibitor of necroptosis [23], was employed before LPS treatment, to explore the contribution of necroptosis to muscle protein degradation.

## 2. Results

### 2.1. LPS Challenge Dynamically Changes Physiological and Biochemical Indexes for Diagnosis of Cachexia in Weanling Pigs

Cachexia is a disease mainly caused by neuroendocrine hormone imbalances or inflammatory factors, leading to muscle protein degradation and increasing energy consumption [1,24,25]. To determine whether the model of LPS-induced cachexia is established, we initially examined the concentrations of several blood markers of cachexia including plasma insulin (INS), growth hormone (GH), cortisol (COR), and glucagon (GLU) [26,27,28]. Compared to the control group, the concentrations of INS and GH in the plasma of LPS-challenged pigs were significantly diminished at 1–4 h (*p* < 0.05) and 1–24 h (*p* < 0.05), respectively (Figure 1A,B). However, the levels of COR and GLU were dramatically increased at 1–12 h (*p* < 0.05) and 2–12 h (*p* < 0.05), respectively (Figure 1C,D).

Next, we further evaluated the occurrence of muscle damage caused by the LPS challenge. Compared with the control group, LPS-challenged pigs displayed muscle-fiber-free nuclei, inflammatory cell infiltration, hemorrhage, membrane rupture, fibroblast hyperplasia, and muscle-fiber dissolution and atrophy (Figure 1E). Muscle ultrastructure analysis showed that the LPS challenge caused mitochondrial cristae shedding and dissolution, muscle-fiber dissolution and fracture, and Z-line fuzziness and distortion at different time points (Figure 1F).

We then investigated muscle protein degradation, a typical symptom of cachexia [1], by measuring the protein and DNA concentrations in the longissimus dorsi muscle. Compared to the control group, protein concentration and protein/DNA ratio of the longissimus dorsi muscle were diminished at 4–24 h (*p <* 0.05), with the lowest point at 12 h after LPS injection (Figure 1G–I). The concentration of DNA tended to decrease from 1–4 h after LPS injection (*p* < 0.10).

### 2.2. LPS Challenge Dynamically Induces Inflammatory Response and Activates Muscle Protein Degradation Signaling Pathway in Weanling Pigs

The inflammatory response is thought to be a major inducer of muscle atrophy in cachexia [29]. To investigate the dynamic effect of LPS challenge on the inflammatory response of pigs, we first measured the mRNA expression of *TNF-α*, *IL-1β*, and *IL-6* in the longissimus dorsi muscle. Compared to the control group, the mRNA abundances of *TNF-α*, *IL-1β*, and *IL-6* in the longissimus dorsi muscle of pigs after LPS injection were significantly elevated at 1 h, 1–24 h, and 1–8 h, respectively (*p <* 0.05), and peaked at 1–2 h after LPS injection (Figure 2A–C).

Pro-inflammatory cytokines induce muscle protein degradation prevailingly by regulating the signaling pathway of Akt/FOXO/UPS [30,31]. To further confirm whether the muscle protein degradation signaling pathway was activated by the LPS challenge in piglets, we assessed the dynamic changes of the key components of the Akt/FOXO/UPS signaling pathway. Specifically, as compared to the control group, the LPS challenge significantly decreased the mRNA level of *Akt* at 24 h (*p* < 0.05) (Figure 2D). The mRNA abundances of *FOXO1*, *MuRF1*, and *MAFbx* were significantly upregulated between 2–12 h, 2–12 h, and 4–24 h after LPS injection, and peaked at 12 h (Figure 2E–G). Similarly, the protein expressions of phosphorylated (p)-Akt and p-FOXO1 and the ratios of p-Akt/total (t)-Akt and p-FOXO1/t-FOXO1 were reduced at 4–24 h, 24 h, 4 h, and 2–12 h, respectively (*p* < 0.05). Injection of LPS upregulated the protein abundances of t-Akt at 2–8 h and t-FOXO1 at 2 and 12 h (*p* < 0.05) (Figure 2H–N).

### 2.3. LPS Challenge Dynamically Induces Muscle Necroptosis in Weanling Pigs

To confirm the involvement of necroptosis in LPS-induced muscle protein degradation, we assessed the mRNA and protein levels of RIP1, RIP3, and MLKL, which are considered to be key signals of necroptosis [17]. Compared to the control group, LPS upregulated the mRNA expression of *RIP1* between 1–24 h (peaked at 12 h), *RIP3* between 1–24 h (peaked at 4 h), and *MLKL* between 1–12 h (peaked at 4 h) (*p* < 0.05) (Figure 3A–C). The protein abundances of t-RIP1 between 1–12 h, p-RIP1 between 1–12 h, t-RIP3 between 1–12 h, p-RIP3 between 2–8 h, t-MLKL between 1–24 h, and p-MLKL between 4–12 h were significantly elevated (*p* < 0.05). The ratios of p-RIP1/t-RIP1 between 1–12 h and p-MLKL/t-MLKL at 24 h were also significantly increased (*p* < 0.05) (Figure 3D–M).

### 2.4. Nec-1 Inhibits Muscle Necroptosis Induced by LPS in Weanling Pigs

Since RIP1 is essential for the induction of canonical necroptosis, we pretreated piglets with Nec-1, an inhibitor of RIP1 kinase activity, to inhibit RIP1 autophosphorylation to block necroptosis. As the results showed (Figure 4A–M), the mRNA expression of *RIP1*, *RIP3*, and *MLKL* and the protein expression of t-RIP1, t-RIP3, and p-MLKL were significantly elevated (*p* < 0.05) after 4 h of LPS injection. Nevertheless, Nec-1 alleviated the increase of the protein abundances of t-RIP1, t-RIP3, and p-MLKL induced by LPS challenge in longissimus dorsi muscle (*p* < 0.05).

### 2.5. Inhibition of Necroptosis by Nec-1 Relieves Muscle Structure Injury, Inflammation, and Protein Degradation Induced by LPS in Weanling Pigs

To further explore whether inhibition of necroptosis alleviated LPS-induced cachexia, we first measured the effect of Nec-1 on blood markers of cachexia. Nec-1 increased the content of GH (*p* < 0.05) and decreased the concentration of COR in plasma (*p* < 0.05) (Figure 5A,B). Compared with the control group, Nec-1 reduced INS level in the blood (*p* < 0.05) (Appendix A). GLU level was increased in LPS + Nec-1 group compared to the LPS group (*p* < 0.05) (Appendix A). In addition, Nec-1 alleviated muscle-fiber-free nuclei, inflammatory cell infiltration, hemorrhage, fascia rupture and shedding, fibroblast proliferation, muscle-fiber atrophy, and other morphological structure damage induced by LPS challenge (Figure 5C). Consistently, LPS treatment diminished the protein concentration and protein/DNA ratio (*p* < 0.05). Nec-1 restored the decrease of protein concentration and protein/DNA ratio induced by LPS challenge to normal levels in the longissimus dorsi muscle (*p* < 0.05) (Figure 5D–F).

We further investigated whether inhibition of necroptosis attenuated inflammatory response and muscle protein degradation. LPS challenge at 4 h upregulated the mRNA expression of *IL-1β* and *IL-6* (*p* < 0.05). As expected, Nec-1 alleviated the increase of *IL-1β* and *IL-6* mRNA abundance induced by the LPS challenge (*p* < 0.05) (Figure 5G–I). In addition, LPS treatment at 4 h elevated *MuRF1* and *MAFbx* mRNA abundance but were alleviated by Nec-1 pretreatment (*p* < 0.05) (Figure 5K–M). Nec-1 pretreatment restored the reduction of the ratio of p-FOXO1/t-FOXO1 to the normal level (*p* < 0.05) (Figure 5N–T).

## 3. Discussion

In recent years, necroptosis, a new type of cell death that is different from traditional cell necrosis and apoptosis, has been confirmed to be involved in physiological and pathological processes of various tissues and organs, such as neurodegenerative disease, intervertebral disc degeneration, and nephritis [32,33,34]. However, whether necroptosis contributes to muscle protein degradation in cachexia remains unclear.

Using LPS-stimulated piglets to establish the model of cachexia, we quantified the concentrations of plasma markers of cachexia, such as INS, GH, COR, and GLU [28]. Various reports have revealed that hormone secretion disorder can be triggered by LPS challenge, specifically manifested as reduced plasma INS and GH concentrations [35,36], and increased plasma COR and GLU concentrations [37,38]. These changes are highly consistent with the physiological changes of cachexia patients. We found that the levels of INS and GH were significantly diminished at 1 h after LPS injection and the concentrations of COR and GLU were dramatically increased at 2–12 h after LPS challenge. In agreement with previously published results of cachexia patients and organisms [26,27,28], the INS and GH levels and the COR and GLU levels were decreased and increased, respectively. Since severe muscle mass loss and protein degradation are manifested in cachexia patients [39], we first explored the dynamic effect of the LPS challenge on muscle damage by observing the morphology and ultrastructure of muscle. As expected, injection of LPS at different time points (1, 2, 4, 8, 12, and 24 h) induced different degrees of muscle injury. We further explored the dynamic effect of the contents of protein and DNA and the ratio of protein/DNA in longissimus dorsi muscle by LPS challenge. The protein/DNA ratio is an effective indicator of muscle protein mass [9]. Our results demonstrated that the content of protein and the ratio of protein/DNA decreased after LPS injection 4–24 h. In agreement with our results, Liu et al. [40] showed that the ratio of protein/DNA was diminished in the longissimus dorsi muscle and gastrocnemius muscle at 4 h after LPS injection. These results indicate that LPS dynamically changes physiological and biochemical indexes for the diagnosis of cachexia in weanling pigs.

Cachexia, a complex and fatal metabolic disease, can cause severe systemic inflammation and obvious muscle atrophy [22]. Proinflammatory cytokines, particularly TNF-α, IL-6, and IL-1β, are considered to be important mediators of cachexia induction [39]. Here, we showed that the mRNA levels of *TNF-α*, *IL-1β*, and *IL-6* increased significantly at 1–8 h after the LPS challenge, and reached the peak value at 1–2 h, which was similar to the results of Valentine et al. [41] and Joseph et al. [42]. These results further proved that cachexia could lead to an inflammatory response in the body. Akt/FOXO/UPS pathway is recognized as the primary mechanism of muscle wasting in cachexia [3,4]. Akt, a key signaling protein, exerts an important role in insulin signaling transduction [43]. It inhibits muscle protein degradation by phosphorylation and inactivation of FOXO [44]. Phosphorylation and inactivation of FOXO lead to transcriptional inhibition of FOXO target genes, i.e., *MAFbx* and *MuRF1*, and consequently result in a reduction in protein degradation. In our study, LPS caused marked but temporary upregulation of the mRNA expression of *FOXO1*, *MuRF1*, and *MAFbx*, peaking at 8–12 h. Similarly, Liu et al. [45] showed that *MuRF1* and *MAFbx* expression was elevated at 4 h after LPS injection in the gastrocnemius muscle of piglets. Furthermore, our data showed that p-Akt protein expression and p-FOXO1/t-FOXO1 ratio were dynamically decreased. These results indicate that LPS-induced proinflammatory cytokines inhibited Akt phosphorylation, thereby reducing FOXO phosphorylation, and consequently increasing the expression of MAFbx and MuRF1 to trigger muscle protein degradation [44,46,47]. According to our data, we demonstrate that LPS induces the release of proinflammatory cytokines and then the activation of muscle protein degradation signals, which are key symptoms of cachexia.

Generally, cell necrosis is closely related to the occurrence of muscle atrophy. Thus, we next elucidated whether the muscle underwent necroptosis after the LPS challenge. RIP1, RIP3, and MLKL are indispensable components that mediated the activation of necroptosis [16,17]. We found that mRNA expression levels of these three key genes were mainly upregulated at 1–12 h and primarily peaked at 4 h. The protein expression displayed an analogous time-dependent increase but principally peaked at 8 h, which coincided with the spatiotemporal differences in the complex gene expression mechanism (involving transcription and translation) in organisms [48]. In agreement with our findings, Morgan et al. [49] reported that the mRNA abundances of *RIP1*, *RIP3*, and *MLKL* were upregulated in the tibialis anterior muscle of mice lacking dystrophin. Our data suggest that LPS induces inflammatory response and muscle protein degradation accompanied by necroptosis of muscle. Interestingly, changes in the major components of necroptosis occurred slightly later than the immediate changes in inflammatory genes and earlier than the larger changes in key genes for muscle protein degradation.

We then pretreated LPS-challenged piglets with Nec-1, a specific inhibitor to block the interaction between RIP1 and RIP3 [50], to investigate whether necroptosis played an important role in muscle protein degradation. We observed that Nec-1 reversed the up-regulation of the protein abundances of t-RIP1, t-RIP3, and p-MLKL in the longissimus dorsi muscle of LPS-treated piglets, suggesting that Nec-1 was efficacious to block necroptosis in the muscle of LPS-challenged pigs. In agreement with our results, Nec-1 also exerted a similar effect in rat models of adjuvant arthritis [51], asthma [52], and hypertrophic scars [53].

Further analysis showed that suppression of necroptosis using Nec-1 ameliorated the change of blood markers (GH and COR) of cachexia by LPS challenge. Nec-1 reduced INS level in the blood. It is well established that INS has the function of reducing blood glucose level but GLU has the function of raising blood glucose level. This may be why the GLU levels was increased in the LPS+Nec-1 group. We also observed that inhibition of necroptosis alleviated muscle morphological damage induced by the LPS challenge. Similar to our research, Morgan et al. [49] showed that RIP3 deficiency significantly reduced inflammatory cell infiltration in muscle fibers and Nec-1s alleviated TNF-α-induced increase of injury and necrosis in C2C12 myoblasts. Remarkably, inhibition of necroptosis by knocking out the *RIP3* gene or using Nec-1s improved muscle-fiber function [49,54]. In addition, our data showed that the up-regulation of muscle *IL-1β* and *IL-6* mRNA abundance induced by LPS were ameliorated by Nec-1. Similarly, recent literature reported that Nec-1 significantly reduced the levels of *TNF-α*, *IL-1β*, and *IL-6* and reduced the severity of arthritis in rats [51,52]. Expectedly, restraining necroptosis alleviated the reduction in protein content, protein/DNA ratio, and p-FOXO1/t-FOXO1 ratio, and the upregulation of *MuRF1* and *MAFbx* mRNA level induced by LPS. Our discovery suggests that necroptosis is involved in muscle inflammation and protein degradation by LPS challenge.

## 4. Materials and Methods

### 4.1. Animals and Experimental Design

Animal housing, feeding management, and all experimental protocols for the piglets used in this study were in compliance with the Animals (Scientific Procedures) Act 1986 under the Home Office Code of Practice. All experiments were performed following the guidelines established by the Animal Care and Use Committee of Wuhan Polytechnic University (Wuhan, China). Sixty-six male weaned piglets (Duroc × Large White × Landrace, 28 ± 3 d of age, initial body weight of 7.1 ± 0.9 kg BW) were provided by Aodeng Agricultural and Animal Husbandry Technology Co., Ltd. (Tianmen, China). All piglets were individually housed in metabolic cages measuring 1.8 m × 1.1 m in length × width and were allowed to eat and drink freely for two weeks. The ambient temperature ranged from 27 °C to 29 °C.

In experiment 1, forty-two weaned piglets were randomly divided into 7 groups (n = 6 piglets/group) according to initial body weight and ancestry, including the control group and six LPS-treated groups. The piglets in six LPS-treated groups received an intraperitoneal injection of *Escherichia coli* LPS (*Escherichia coli* serotype O55: B5, Sigma Chemical Inc., St. Louis, MO, USA) at the dose of 100 μg/kg BW, and then were humanly killed at 1, 2, 4, 8, 12, and 24 h, respectively. The piglets in the control group were injected with the same amount of sterile saline and slaughtered immediately (0 h).

In experiment 2, twenty-four weaned piglets were randomly allocated to 4 groups (n = 6 piglets/group) according to initial body weight and ancestry with a 2 × 2 factorial arrangement of treatments. The pigs were pretreated with Nec-1 (MedChem Express, Monmouth Junction, NJ, USA) at 1.0 mg/kg BW or the equal quantity of 2% dimethylsulphoxide (DMSO, the solvent of Nec-1) 30 min earlier than the intraperitoneal injection of LPS or sterile saline, and were slaughtered at 4 h after injection of LPS or sterile saline.

### 4.2. Blood and Muscle Sample Collection

Blood samples were collected initially into 10 mL vacuum tubes by the precaval vein, and plasma was obtained by centrifugation at 3500× *g* and 4 °C for 10 min. Plasma samples were stored at −20 °C until analysis. After blood collection, the pigs were slaughtered under anesthesia with an intramuscular administration of sodium pentobarbital (80 mg/kg BW). Two patch muscles (3 mm^3^ and 1 cm^3^) were harvested from the longissimus dorsi muscle. The smaller piece of muscle was rinsed with phosphate solution at 4 °C, and fixed with 2.5% glutaraldehyde stored at 4 °C for ultrastructural analysis. Another bigger piece of muscle was fixed with 4% paraformaldehyde for morphological analysis. Furthermore, a portion of the remaining longissimus dorsi muscle was removed and frozen in liquid nitrogen immediately, then stored at −80 °C for further analysis.

### 4.3. Plasma Hormone Measurement

The concentrations of INS, GH, COR, and GLU in the plasma were measured by commercial ^125^I RIA assay kits (Beijing North Institute of Biological Technology Co., Ltd., Beijing, China). All experimental processes were conducted in accordance with the manufacturer’s protocols.

### 4.4. Muscle Morphologic Analysis

Muscle segments (1 cm^3^) fixed with 4% paraformaldehyde-phosphate buffer were dehydrated with a series of increasing concentration gradient of ethanol, cleared with xylene, embedded with Paraffin wax, sectioned (5 μm) and stained with hematoxylin and eosin. Specifically, slices of each sample were made respectively along the transverse and longitudinal sides of the muscle-fiber texture. Morphological observation of the longissimus dorsi muscle was conducted at 400× magnification of a light microscope.

### 4.5. Muscle Ultrastructural Observation

The smaller pieces of muscle (3 mm^3^) from each pig were fixed with 2.5% glutaraldehyde, and subsequently fixed in 1% osmium tetroxide. The muscle samples were then dehydrated with graded ethanol and acetones, permeated in a mixture of acetone and epoxy resin, embedded in Epon 812 (Eimicon, Shanghai, China), cut into ultrathin sections (70 nm), and stained with uranyl acetate and lead citrate. Ultrastructural analysis of muscle was evaluated by using a transmission electron microscope (TEM) (Tecnai, FEI, Hillsboro, OR, USA) at an accelerating voltage of 200 kV and a magnification of 6000 by a seasoned pathologist blinded manner.

### 4.6. Muscle Protein and DNA Concentration Determination

Muscle samples of 100 mg were weighed and added to pre-cooled saline at a 1:10 (*w*/*v*) ratio and then homogenized with a homogenizer (PT-3100D, Kinematica, Malters, Switzerland) to prepare 10% tissue homogenate for further analysis. The concentration of muscle protein was determined using Coomassie brilliant blue kit (Nanjing Jiancheng Bioengineering Institute, Nanjing, China). Muscle DNA content was measured by UV spectrophotometry method [55].

### 4.7. mRNA Abundance Analysis

An appropriate muscle sample (50−70 mg) was added into the enzyme-free 2 mL eppendorf tube which contained 1 mL Trizol reagent (TaKaRa Biotechnology Co., Dalian, China) to obtain total RNA. RNA quality was evaluated by agarose gel electrophoresis, and its quantity was detected by using a Nanodrop 2000 Spectrophotometer (Thermo Scientific, Wilmington, DE, USA). cDNA synthesis and quantitative real-time PCR were executed using PrimeScript^®^ RT Kit (TaKaRa Biotechnology Co., Ltd., Dalian, China) and SYBR^®^ Premix Ex Taq™ (Tli RNaseH Plus) qPCR kit (TaKaRa Biotechnology Co., Ltd., Dalian, China), respectively. The PCR cycling conditions included denaturation at 95 °C for 30 s, followed by 40 cycles of annealing at 95 °C for 5 s and extension at 60 °C for 34 s. The primer pairs designed with Primer Premier 6.0 are shown in Table 1. Relative mRNA expression of each target gene was normalized to the expression of the housekeeping gene (GAPDH), and was calculated according to the 2^−ΔΔCT^ method [56].

### 4.8. Protein Expression Analysis

Total proteins were extracted from muscle using a commercial total protein extraction kit (#KGP250/KGP2100, Jiangsu KeyGEN BioTECH Corp., Ltd., Nanjing, China) according to the operation instruction. Protein samples were separated by electrophoresis on 12% sodium dodecyl sulfate-polyacrylamide gel, transferred onto polyvinylidene difluoride membranes, blocked with 3% bovine serum albumin in TBS/Tween-20 buffer, and incubated with primary antibodies and then with secondary antibodies. The information of antibodies used in the Western blot is shown in Table 2. The protein content of the band was measured by using GeneTools software (Syngene, Frederick, MD, USA). The relative concentration of each target protein was indicated as the ratio of target protein/GAPDH protein. In addition, the protein abundance of the phosphorylated form of the target protein was also normalized to its total protein content.

### 4.9. Statistical Analysis

For comparison between the two groups, the data were analyzed using Student’s *t*-test. For multiple comparisons, a one-way ANOVA test was performed using the general linear model procedures of Statistical Analysis System (SAS Inst. Inc., Cary, NC, USA), followed by Duncan’s multiple comparison tests for post hoc testing. The experimental data were expressed as means ± SEM. The statistical significance level of data was set at *p* ≤ 0.05.

## 5. Conclusions

Collectively, we demonstrate that necroptosis plays an important role in muscle protein degradation in LPS-induced cachexia. More specifically, muscle inflammation, protein degradation, and necroptosis occur in the LPS-induced cachexia model of weaned piglets. Inhibition of necroptosis with Nec-1 alleviates muscle inflammation and protein degradation. This suggests that inhibitors of necroptosis may be promising new drugs for treating muscle protein degradation caused by cachexia. Future research is warranted to elucidate the specific mechanism of necroptosis mediating muscle protein degradation. Moreover, the study of pretreatment with Nec-1s, a more specific inhibitor than Nec-1, is worth pursuing in the future to exclude interference from other modes of cell death.

## Figures and Tables

**Figure 1 ijms-24-10923-f001:**
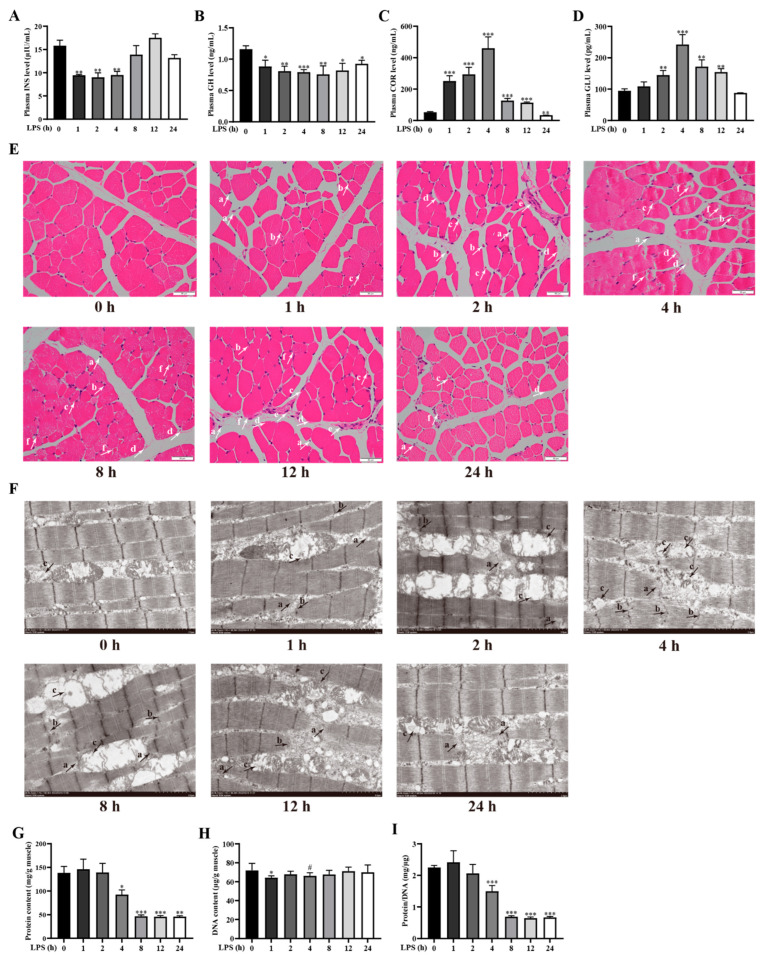
LPS challenge dynamically changes physiological and biochemical indexes for the diagnosis of cachexia in weanling pigs. Blood and longissimus dorsi muscle samples were collected from the control group (0 h after injection of saline) and LPS-treated groups (1, 2, 4, 8, 12, and 24 h after injection of LPS) (**A**–**D**) The concentrations of INS, GH, COR, and GLU in the plasma. (**E**) Representative morphological characteristics of H&E staining of longissimus dorsi muscle. Arrows indicate muscle-fiber-free nuclei (a), inflammatory cell infiltration (b), hemorrhage (c), membrane rupture (d), fibroblast hyperplasia (e), and muscle-fiber dissolution and atrophy (f). Original magnification: 400×, scale bars = 50 μm. (**F**) Representative ultrastructure of longissimus dorsi muscle observed by using TEM. Arrows indicate mitochondrial cristae shedding and dissolution (a), muscle-fiber dissolution and fracture (b), and Z-line fuzziness and distortion (c). Magnification: 6000×, scale bars = 2 μm. (**G**–**I**) Protein and DNA concentrations and protein/DNA ratio in muscle. Data are presented as means ± standard errors of the mean (SEM), *n* = 6. *** *p* < 0.001, ** *p* < 0.01, * *p < 0.05*, and # 0.05 < *p* < 0.10, different from the control group (0 h).

**Figure 2 ijms-24-10923-f002:**
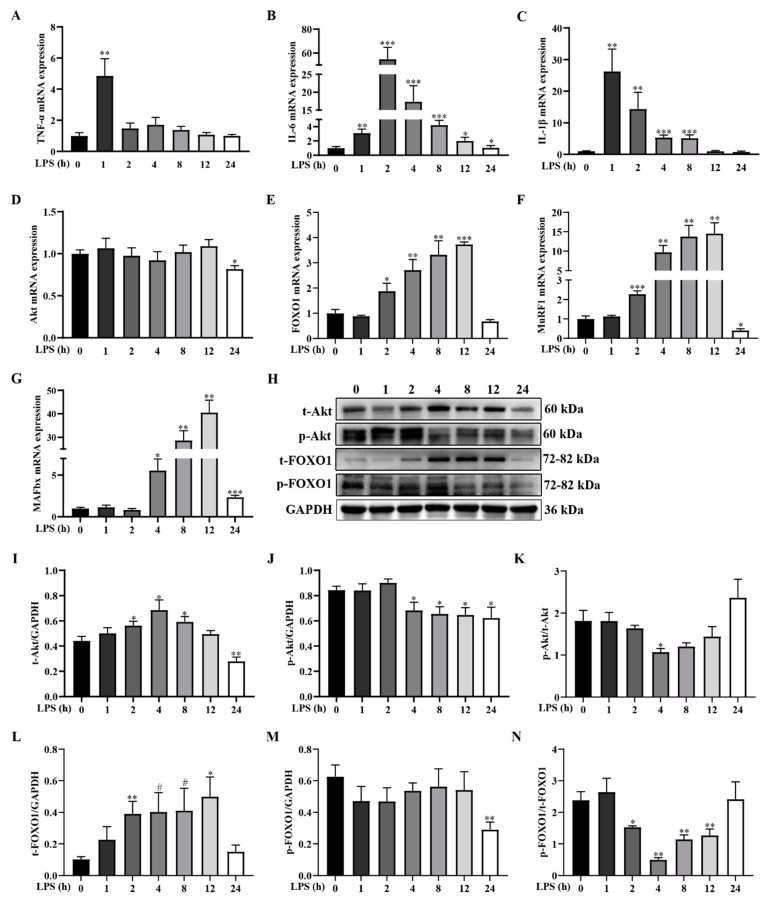
LPS challenge dynamically induces inflammatory response and activates muscle protein degradation signaling pathway in weanling pigs. Longissimus dorsi muscle samples were collected from the control group (0 h after injection of saline) and LPS-treated groups (1, 2, 4, 8, 12, and 24 h after injection of LPS). (**A**–**G**) mRNA levels of pro-inflammatory cytokines (including *TNF-α*, *IL-1β*, and *IL-6*) and the key components of muscle protein degradation signaling pathway (including *Akt*, *FOXO1*, *MuRF1*, and *MAFbx*). (**H**) Representative image of t-Akt, p-Akt, t-FOXO1, and p-FOXO1 protein expression. (**I**–**N**) Protein expression of t-Akt, p-Akt, t-FOXO1, and p-FOXO1. Data are presented as means ± SEM, *n* = 6. *** *p* < 0.001, ** *p* < 0.01, * *p < 0.05*, and # 0.05 < *p* < 0.10, different from the control group (0 h).

**Figure 3 ijms-24-10923-f003:**
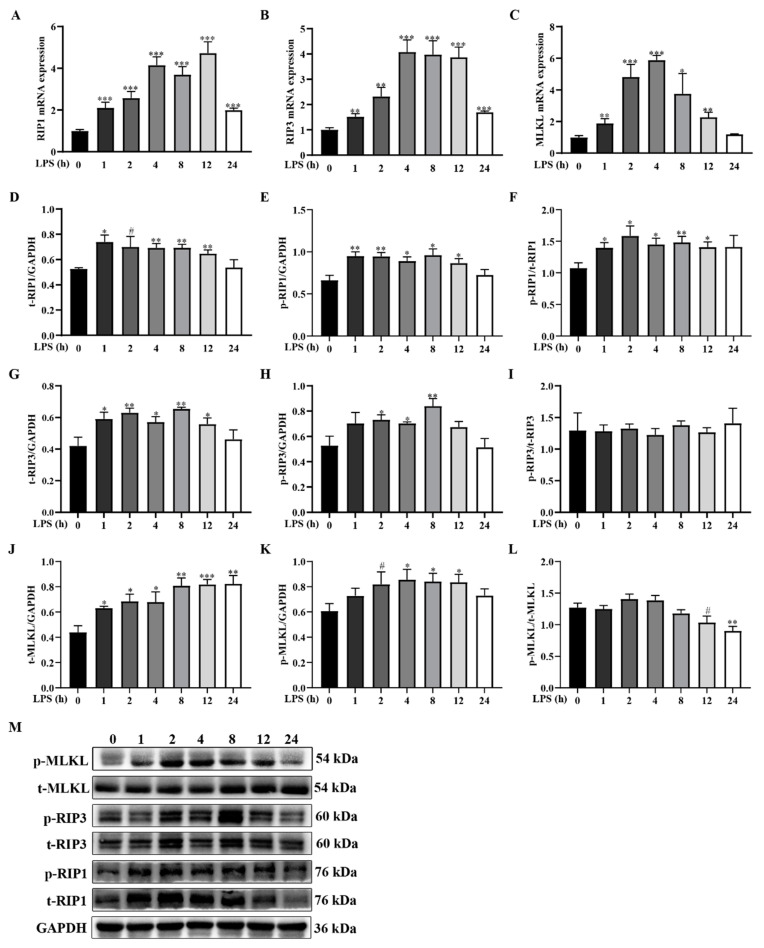
LPS challenge dynamically induces muscle necroptosis in weanling pigs. Longissimus dorsi muscle samples were collected from the control group (0 h after injection of saline) and LPS-treated groups (1, 2, 4, 8,12, and 24 h after injection of LPS). (**A**–**C**) mRNA levels of *RIP1*, *RIP3*, and *MLKL*. (**D**–**M**) Protein expression of t-RIP1, p-RIP1, t-RIP3, p-RIP3, t-MLKL, and p-MLKL. Data are presented as means ± SEM, *n* = 6. *** *p* < 0.001, ** *p* < 0.01, * *p* < 0.05, and # 0.05 < *p* < 0.10, different from the control group (0 h).

**Figure 4 ijms-24-10923-f004:**
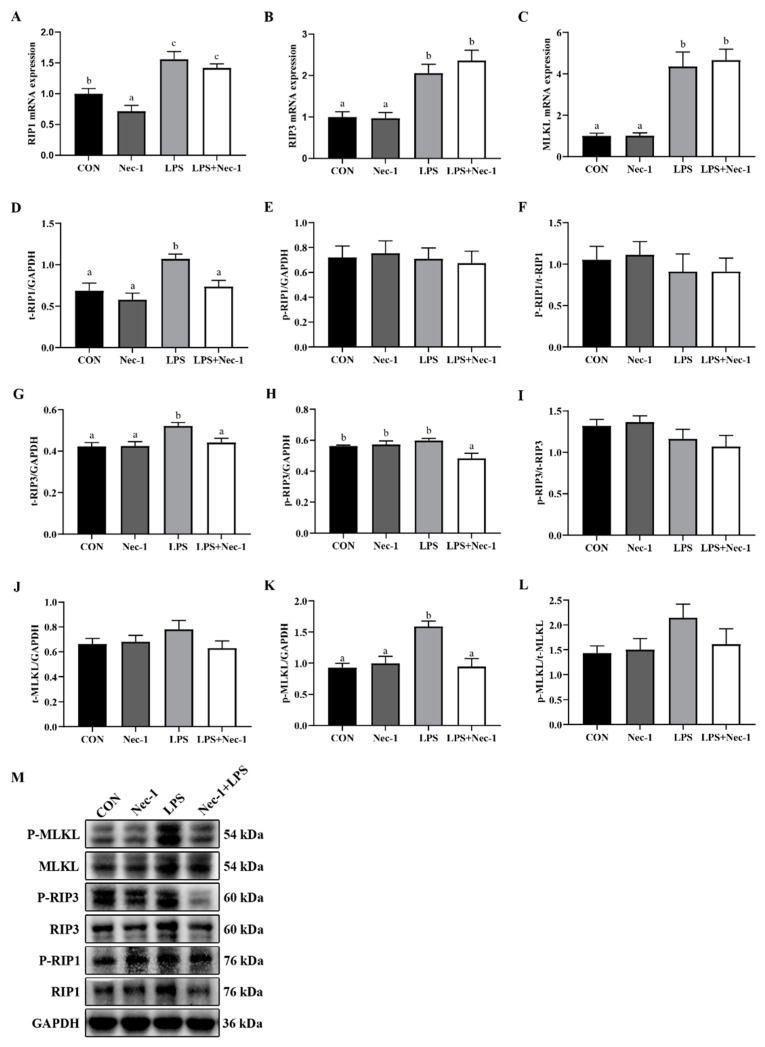
Nec-1 inhibits muscle necroptosis induced by LPS in weanling pigs. The piglets were pretreated with Nec-1 or DMSO for 30 min and then injected with saline or LPS for 4 h. (**A**–**C**) mRNA levels of *RIP1*, *RIP3*, and *MLKL*. (**D**–**M**) Protein expression of t-RIP1, p-RIP1, t-RIP3, p-RIP3, t-MLKL, and p-MLKL. Data are presented as the means ± SEM, *n* = 6. ^abc^ means without a common letter differ (*p* < 0.05).

**Figure 5 ijms-24-10923-f005:**
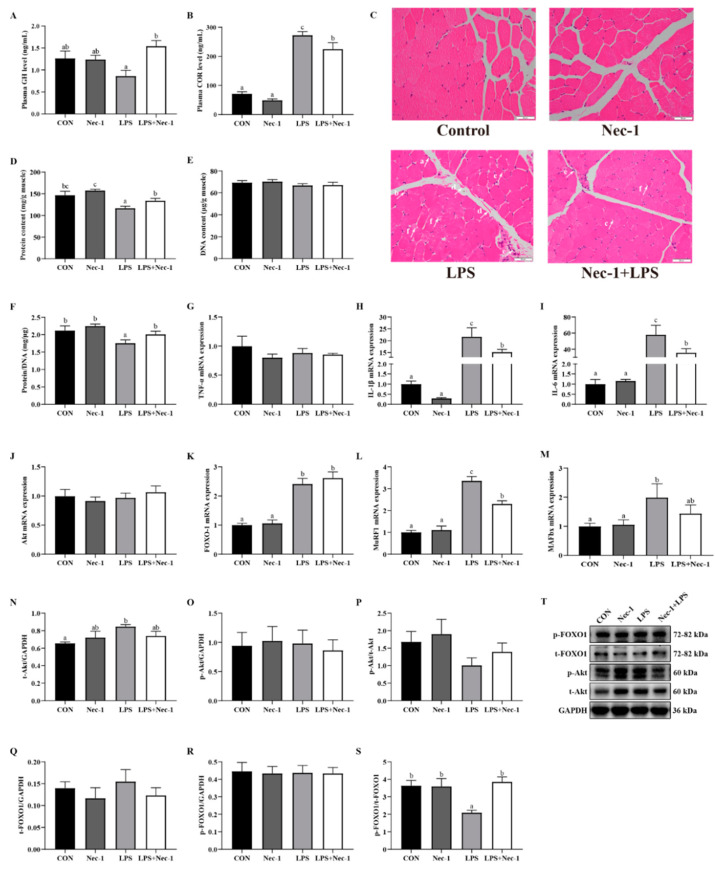
Inhibition of necroptosis by Nec-1 relieves muscle structure injury, inflammation, and protein degradation induced by LPS in weanling pigs. The piglets were pretreated with Nec-1 or DMSO for 30 min and then injected with saline or LPS for 4 h. (**A**,**B**) The concentrations of GH and COR in the plasma. (**C**) Representative morphological characteristics of H&E staining of longissimus dorsi muscle. Arrows indicate muscle-fiber-free nuclei (a), inflammatory cell infiltration (b), hemorrhage (c), membrane rupture (d), fibroblast hyperplasia (e), and muscle-fiber dissolution and atrophy (f). Original magnification: 400×, scale bars = 50 μm. (**D**–**F**) Protein and DNA concentrations and protein/DNA ratio in muscle. (**G**–**M**) mRNA levels of pro-inflammatory cytokines (including *TNF-α*, *IL-1β*, and *IL-6*) and the key components of muscle protein degradation signaling pathway (including *Akt*, *FOXO1*, *MuRF1*, and *MAFbx*). (**N**–**T**) Protein expression of t-Akt, p-Akt, t-FOXO1, and p-FOXO1. Data are presented as the means ± SEM, *n* = 6. ^abc^ means without a common letter differ (*p* < 0.05).

**Table 1 ijms-24-10923-t001:** Primer sequences used in real-time PCR.

Gene	Primer Sequence	Amplification Length	SERIAL NUMBER
*TNF-α*	F: TCCAATGGCAGAGTGGGTATG	67	NM_214022.1
	R: AGCTGGTTGTCTTTCAGCTTCAC		
*IL-1β*	F: GCTAACTACGGTGACAACAATAATG	186	NM_214055.1
	R: CTTCTCCACTGCCACGATGA		
*IL-6*	F: AAGGTGATGCCACCTCAGAC	151	JQ839263.1
	R: TCTGCCAGTACCTCCTTGCT		
*Akt*	F: GAAGAAGGAGGTCATCGT	178	NM_001159776.1
	R: GGACAGGTGGAAGAAGAG		
*FOXO-1*	F: TTCACCAGGCACCATCAT	236	NM_214014.2
	R: GAGGAGAGTCGGAAGTAAGT		
*MuFR1*	F: ATGGAGAACCTGGAGAAGCA	219	FJ905227.1
	R: ACGGTCCATGATCACCTCAT		
*MAFbx*	F: TCACAGCTCACATCCCTGAG	167	NM_001044588.1
	R: GACTTGCCGACTCTCTGGAC		
*RIP1*	F: ACATCCTGTACGGCAACTCT	175	XM_005665538.2
	R: CGGGTCCAGGTGTTTATCC		
*RIP3*	F: CTTGTTGTCTGTCCGTGAGC	238	XM_001927424.3
	R: GAGGAGGTTGGGCTGTTGA		
*MLKL*	F: TCTCGCTGCTGCTTCA	105	XM_013998184.1
	R: CTCGCTTGTCTTCCTCTG		
*GAPDH*	F: CGTCCCTGAGACACGATGGT	194	AF017079.1
	R: GCCTTGACTGTGCCGTGGAAT		

*TNF*, Tumor necrosis factor; *IL*, Interleukin; *Akt*, Serine/threonine kinase; *FOXO1*, Forkhead box O 1; *MuRF1*, Muscle RING finger 1; *MAFbx*, Muscle atrophy F-box; *RIP*, Receptor interacting protein kinase; *MLKL*, Mixed lineage kinase domain-like protein; *GAPDH*, Glyceraldehyde-3-phosphate dehydrogenase.

**Table 2 ijms-24-10923-t002:** The information of antibodies used in Western blot.

Antibodies (Host)	Catalog Number	Manufacturers	Dilution Ratio
RIP1 (rabbit)	LS-B8214	LifeSpan BioSciences (Lynnwood, WA, USA)	1:1000
p-RIP1 (rabbit)	91702S	Cell Signaling Technology (Danvers, MA, USA)	1:1000
RIP3 (rabbit)	SC-135170	Santa Cruz Biotechnology (Dallas, TX, USA)	1:1000
p-RIP1 (rabbit)	91702S	Cell Signaling Technology	1:1000
MLKL (rabbit)	37705S	Cell Signaling Technology	1:1000
p-MLKL (rabbit)	62233S	Cell Signaling Technology	1:1000
Akt (rabbit)	9272S	Cell Signaling Technology	1:1000
p-Akt (rabbit)	9271S	Cell Signaling Technology	1:1000
FOXO1 (rabbit)	9454S	Cell Signaling Technology	1:1000
p-FOXO1 (rabbit)	9464S	Cell Signaling Technology	1:1000
GAPDH (mouse)	FD0063	Hangzhou Fude Biological Technology (Hangzhou, China)	1:5000
IgG-HRP (goat anti-rabbit)	ANT020	Wuhan Antejie Biotechnology(Wuhan, China)	1:5000
IgG-HRP (goat anti-mouse)	ANT019	Wuhan Antejie Biotechnology	1:5000

## Data Availability

The data used to support the findings of this study are included in the article.

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
