# Peer review of "Necroptosis Mediates Muscle Protein Degradation in a Cachexia Model of Weanling Pig with Lipopolysaccharide Challenge"

_ijms, 2023, doi:10.3390/ijms241310923_

Round 1

Reviewer 1 Report

In this study the authors demonstrate the possible role of necroptosis in the pathomechanism of cachexia by in vivo experimental models. The manuscript is well-written and readable, measurements are comprehensive.

In my opinion, the main shortcoming of the study is the justificatino of the chosen animal model. It is not clear, whether LPS treatment is a generally accepted model for muscle protein degradation and/or cachexia. Disorders, described in the introduction like cancer or chemo-associated cachexia has special in vivo experimental models. Intraperitoneal injection of LPS models rather the mechanism of sepsis, inducing severe systemic inflammation. LPS can directly increase the production of inflammatory cytokines as TNF-a, IL-1ß, IL-6 via TLR-related signaling pathways. In addition the chosen model is ultra short (few hours), while cachexia is a consequence of chronic diseases and other processes.

Beside the inflammatory cytokines, it is known that LPS can directly influence the expression INS, GH, COR and GLU in vitro. Authors should mention and discuss the previous known literary data in the discussion.

Other comments:

abstract

- Authors should use an other adjective instead of "new form". Necroptosis is an actively researched topic since the early 2000s.

results

- Abbreviations should be defined at their first appearance (e.g. page 2: INS, GH, COR, GLU, stc.).

- INS, GH, COR, GLU should be defined as markers or variable parameters in cachexia instead of biomarkers, as they are not only cachexia-specific molecules

- in Fig 2/K the authors show that LPS treatment induce decreased Akt-phosphorilation, which is the opposite of the known effect of LPS from other studies. Results (not only these) should be discuss and place in the literary data.

- in Fig 5/A and B, GH and COR levels of pigs are illustrated. In the pilot experiment, INS and GLU levels were also altered at 4h (Fig 1/A-D). How did these values change in the second experiment?

- in Fig 5/D the authors illustrate the protein content of muscle tissues in mg/g dimension. In the pilot experiment (Fig 1/G), the protein content of control group was 3-fold higher (150 +/- 10 ) compared to the values derived from the 2nd experiment (50 +/- 5). What could be the explanation of this phenomenon?

Reviewer 2 Report

Introduction.

Please define clearly the objectives of the study in a distinct paragraph.

M & M

Please describe all the details of the methodology used to allocate animals into groups.

Table 1. All the details of PCR should be mentioned (temperature, product size etc.) The table to be included as appendix.

4.8. Details of antibodies used to be included in a table in appendix.

Results

Please colorize all graphs and please maintain consistency of colors in the graphs.

Discussion

This is rather short and shallow.

The authors do not explain adequately their findings. The section must be extended and further relevant references must be discussed.

Extensive editing of English language required.

Round 2

Reviewer 2 Report

The clinical significance of the findings must be highlighted in a final paragraph in the Discussion section.

Minor editing of English language required.
